# Capture-Aware Dense Tag Identification Using RFID Systems in Vehicular Networks

**DOI:** 10.3390/s23156792

**Published:** 2023-07-29

**Authors:** Weijian Xu, Zhongzhe Song, Yanglong Sun, Yang Wang, Lianyou Lai

**Affiliations:** 1School of Ocean Information Engineering, Jimei University, Xiamen 361000, China; xwjxwj@jmu.edu.cn (W.X.); 202111810010@jmu.edu.cn (Z.S.); 2Navigation Institute, Jimei University, Xiamen 361000, China; syl@jmu.edu.cn; 3School of Informatics, Xiamen University, Xiamen 361000, China; yangwang@stu.xmu.edu.cn

**Keywords:** vehicular networks, RFID, tag identification, capture effect

## Abstract

Passive radio-frequency identification (RFID) systems have been widely applied in different fields, including vehicle access control, industrial production, and logistics tracking, due to their ability to improve work quality and management efficiency at a low cost. However, in an intersection situation where tags are densely distributed with vehicle gathering, the wireless channel becomes extremely complex, and the readers on the roadside may only decode the information from the strongest tag due to the capture effect, resulting in tag misses and considerably reducing the performance of tag identification. Therefore, it is crucial to design an efficient and reliable tag-identification algorithm in order to obtain information from vehicle and cargo tags under adverse traffic conditions, ensuring the successful application of RFID technology. In this paper, we first establish a Nakagami-*m* distributed channel capture model for RFID systems and provide an expression for the capture probability, where each channel is modeled as any relevant Nakagami-*m* distribution. Secondly, an advanced capture-aware tag-estimation scheme is proposed. Finally, extensive Monte Carlo simulations show that the proposed algorithm has strong adaptability to circumstances for capturing under-fading channels and outperforms the existing algorithms in terms of complexity and reliability of tag identification.

## 1. Introduction

The vehicle networking industry is booming due to the rapid development of emerging technologies such as edge computing, wireless communication, and artificial intelligence [1,2,3,4,5,6,7]. Compared to visual inspection technology, which relies on “seeing” assets, RFID solutions rely on “listening” by using ultra-high-frequency (UHF) signals to remotely interrogate RFID tags attached to or embedded in objects, which greatly improves the efficiency of traffic control systems, including electronic toll collection, cargo inspection, and vehicle entry control, among other applications [8,9].

A complete UHF RFID system consists of readers, UHF tags, and back-end servers [10]. Each tag has a globally unique electronic product code (EPC). However, collisions may occur when multiple vehicles or cargo tags use backward scattering modulation to respond to readers simultaneously on the same cascaded channel [11,12]. Therefore, the RFID system needs to effectively coordinate the network and employ anti-collision algorithms. Collisions between readers will also certainly occur in the application system for intensive readers. In particular, with the rise of reinforcement learning [13,14,15,16], it is necessary for RFID systems that a multi-reader anti-collision algorithm based on reinforcement learning is used. However, this paper focuses on the field of identifying tags for a single reader in densely distributed vehicular networks.

Anti-collision algorithms are divided into two categories: the tree-based algorithm [17,18,19] (including query tree and tree splitting) and the ALOHA-based algorithm [20,21,22,23]. The former requires high computing costs when solving the problem of tag starvation. In addition, when the length of the tag ID is long, the tree-based algorithm causes frequent time slot collision, resulting in a large identification delay [20]. In contrast, the ALOHA-based algorithm guarantees equal opportunity for tags in random access, especially in table-dense scenarios with better adaptability [21]. It is favored by most RFID vendors due to its compliance with EPC Class 1 Gen2 UHF standards [24]. The ALOHA-based tag-identification algorithm is designed for rapid vehicle identification on expressways [22,23]. However, the aforementioned works do not comprehensively take into account actual difficulties and simply concentrate on tag recognition in ideal circumstances, neglecting the capture effect.

In Figure 1, as roadside observers, readers can swiftly discern multiple densely clustered tags within each vehicle as they congregate at intersections and obtain reliable target information by using RFID technology [25]. However, in wireless communication, the capture effect is a ubiquitous phenomenon [26]. For instance, in scenarios of checkpoints and intersections, tag density increases significantly with vehicle aggregation, which causes a complex RFID channel condition and frequent timeslots collision. The impact of the capture effect on the system is amplified as the number of collision timeslots increases. Furthermore, for the densely distributed tags, identification delay is a crucial issue for vehicular networks. Thus, it is necessary to mitigate the capture effect under collision timeslots to reduce identification delay and improve system efficiency.

An analytic model for the capture effect categorizes tags based on signal strength [27]. Subsequently, to study the relationship between frame length and the number of tags that can be identified under unreliable channels, some work has focused on improving the efficiency of tag identification through capture-aware anti-collision schemes on the reader side [28,29,30,31]. The capture-aware backlog-estimation algorithm (CMEBE) estimates tag population and the probability of the capture effect with two-dimensional searches for minimum values [28]. However, it is difficult for the the CMEBE algorithm to guarantee identification performance under a large-scale tag environment. A capture-aware estimation (CAE) algorithm also simultaneously estimates the tag population and the probability of capture effect based on the number of idle slots in a frame [29]. Although it has low computational complexity, the algorithm fails in tag identification when the number of tags is much larger than the frame length. Minimum mean square error (MMSE) and Bayesian mean square estimation (CBMS) are proposed respectively in Ref. [30,31] for large-scale tag identification scenarios. However, the computational cost is increased because the algorithm uses a 2D search. Although researchers have begun to study the effect of capture on the system identification rate of RFID systems, this study is still in its infancy. They only give a fixed value of the capture probability, and the estimation performance and computational cost need to be improved.

Some researchers have explored the capture effect by establishing an RFID model through the stochastic distribution of tag positions, using a uniform or sigmoid-shaped distribution function for distance. In Ref. [32], the authors discussed an RFID model related to the capture effect under unreliable fading channel conditions. In Ref. [33,34], the authors provided a collision-avoidance algorithm for mobile tags under complex channel conditions. However, these studies fail to account for the impact of the capture effect on system identification efficiency in scenarios where tags are densely distributed within a specific area. In fact, due to the dense concentration of numerous tags in a particular region, the wireless channel becomes highly complex during tag identification [35,36], significantly affecting the identification efficiency of the system. Relying on path-loss analysis to determine capture probability is insufficient. In Ref. [37], the authors discussed the advantages of the RFID system in traffic recognition compared to visual detection under dense tags and proposed a traffic sign inventory-management system. However, it does not further study the reader’s recognition performance in the context of dense tags. Therefore, it is unclear how the capture effect behaves when tags are located at the same distance or densely distributed. In this work, we construct a channel model to explore and analyze the capture probability of dense tag distributions and their influence on the performance of tag-identification algorithms. The contributions of this paper are as follows:A fading channel-capture model is established to analyze the capture probability in different channel environments, and the closed expression of the Nakagami-*m* fading channel capture probability is derived.We propose an advanced capture-aware estimation algorithm that quickly adjusts the initial frame length through the first few timeslots in a frame, reduces the delay caused by the lack of prior knowledge of the number of tags, and improves the estimation performance of both tags and capture probabilities.Considering the capture effect in fading channels and the duration of the slots, this paper dynamically adjusts the size of the next frame by combining the estimate of the number of tags and the capture probability, thus greatly improving the tag-identification rate. Compared with other excellent algorithms, the estimation method proposed in this paper shows better identification performance.

The remainder of the paper is organized as follows. Section 2 introduces the background, Section 3 establishes the system model, Section 4 shows the proposed estimation algorithm and the optimal frame length strategy. The numerical and analytical results are presented in Section 5, and Section 6 concludes the paper with an overview of some crucial points.

## 2. Background

### 2.1. Brief Analysis of ALOHA-Based Anti-Collision Algorithm

Each time slot in the UHF RFID system is expected to have three states: collision, successful, and idle. Based on statistics, the probability of collision timeslots, successful timeslots, and idle timeslots in the ALOHA algorithm follows a binomial distribution
(1)$$p\left(r\right)=\left(\begin{array}{c} n\\ r \end{array}\right)\left(\frac{1}{l}\right){\left(1-\frac{1}{l}\right)}^{n-r}$$
where *n* is the number of tags in the system, *r* is the number of tags in the same timeslot, and *l* is the size of the frame length. Therefore, the number of idle timeslots is $N_{i}=lp\left(0\right)$, the number of success timeslots is $N_{s}=lp\left(1\right)$, and the number of collision timeslots can be expressed as $N_{c}=l-N_{i}-N_{s}$. Compared to the basic frame-slotted ALOHA (FSA) with a fixed frame length, the dynamic frame-length adjustment algorithm (DFSA) dynamically adjusts the size of the next frame according to the number of tags, which makes tag identification more stable [38]. Therefore, the next frame length needs to be determined by the backlog of the number of unidentified tags, which requires the system to estimate it in advance. At present, the anti-collision scheme in the two major standards, EPC C1 G2 standard and ISO18000-6C standard TypeA [39], is based on the DFSA.

### 2.2. The Entire System under the Capture Effect

In fact, due to the near–far effect and the random fluctuation of the received signal power caused by fading and shading, the probability of a packet being successfully received is significant even when multiple packets are transmitted at the same timeslot, which is called the capture effect [40]. An example of detection results from ALOHA, when the capture effect occurs, can be seen in Figure 2, where the collision state can transform into a successful state with a certain probability because of the capture effect. Therefore, the application of traditional tag estimation and its optimal frame configuration is challenging due to inaccurate backlog estimation, which in turn poses an obstacle to the identification performance of UHF RFID tags [34].

Under the capture effect, if the capture probability is α, then the numbers of idle timeslots, successful timeslots, and collision timeslots are as follows
(2)$$\begin{array}{c} N_{i}^{cap}=N_{i}\\ N_{s}^{cap}=N_{s}+N_{c}\alpha \\ N_{c}^{cap}=N_{c}-N_{c}\alpha \end{array}.$$

If the duration of all three states is equal, the system throughput can be expressed as
(3)$$\eta =\frac{N_{s}^{cap}}{N_{i}^{cap}+N_{s}^{cap}+N_{c}^{cap}}=\frac{n}{l}{\left(1-\frac{1}{l}\right)}^{n-1}+\left(1-\left[{\left(1-\frac{1}{l}\right)}^{n}\left(1+\frac{n}{l-1}\right)\right]\right)\alpha .$$

Figure 3 presents the expectation of system throughput under the non-capture effect and capture effect when a fixed frame length and the DFSA are adopted. The results show that the capture effect can enhance system throughput. This work has defaulted to the traditional frame-length adjustment scheme for DFSA, which sets the next frame length to an unrecognized number of tags. The first curve in the legend using DFSA converges to the optimal value of FSA under non-capture effects. The second curve cannot converge to the optimal value of FSA under the capture effect, which indicates that the next frame length cannot simply be set to an unidentified number of tags under the capture effect. Therefore, the adjustment of the next frame length under the capture effect needs further research, which will be discussed in Section 4.

## 3. Capture Model of High-Density Tag Distribution

To analyze the capture effect in densely distributed tags and its impact on anti-collision algorithms, this work models this complex channel and uses a fading channel to simulate interference and the attenuation of RF signals during transmission.

From Figure 4, the reader communicates with *n* tags densely stored in each container through one channel, which is characterized by its connection to two channels, namely the forward link and the reverse link. The forward channel coefficient $h_{f,D}$ describes the signal propagation from the reader to the tag, while the reverse channel coefficient $h_{b,D}$ describes the signal propagation from the tag to the reader after scattering.

In a fading channel, signal strength varies randomly due to multipath propagation, shadowing effects, and other factors. Thus, this work takes the strength of the signal received between the tags as a sample of the signal during that period.

In Section 2, this work concludes that the capture effect occurs during the collision timeslot. Therefore, this work assumes that there are *s* ($s\in \left[2,s_{\max}\right]$) tags in a collision timeslot, where $s_{max}$ is the maximum number of tags. The *t*-th tag is successfully captured, and the remaining s−1 tags are interference tags or noise for the RFID capture model, where tags are located at the same distance. Then, the probability of the *t*-th tag being captured can be defined as
(4)$$M_{s}\left(q\right)=Pr\left\{P_{R_{t}}>qP_{R_{IS}}\right\}$$
where $P_{R_{t}}$ is the received power from the *t*-th tag, $P_{R_{IS}}=\sum_{k=1}^{s-1}P_{Rd,rx,k}$ is the hidden power from other tags, and *q* represents the power ratio threshold, which represents the minimum carrier-to-Interference Ratio (CIR) required by the reader to receive signals successfully. This CIR is primarily determined by the modulation mode and coding scheme used by the reader. The capture probability can then be written as [41]
(5)$$\alpha =s\int_{0}^{\infty }f_{p_{t}}\left(p_{t}\right)M_{s}\left(q\right)dp_{t}=s\int_{0}^{\infty }f_{p_{t}}\left(p_{t}\right)\left[\int_{0}^{p_{t}/q}f_{p_{s-1}}\left(p_{s-1}\right)dp_{s-1}\right]dp_{t}$$
where $f_{p_{s-1}}\left(p_{s-1}\right)$ is the pdf (for the power) that results from the convolution of s−1 non-central Chi-square pdf’s.

According to the expression [34] for the received power of a reader in free space, assuming constant transmit power and carrier frequency of the reader, the received power is inversely proportional to distance *d*. However, when tags are densely distributed and located very close to each other, the distance between each tag and the reader is also close. Assuming equal reflected power from each interference tag, this work uses the default that the power of other interference signals received by the reader in each collision slot is approximately equal. In addition, these articles [42,43,44] assume that the wireless channel of the RFID system experiences Rayleigh fading and Rician fading. However, Nakagami-*m* is a multi-application model that is suitable for various channel conditions [45], and its distribution is closer to experimental data in various wireless communication environments [46]. Therefore, this work considers a Nakagami-*m* fading channel, ignoring the effect of propagation path loss and assuming that the mean received power of all interfering signals is the same. It is worth noting that the Nakagami-*m* fading channel model is only used to simulate the noise and attenuation effects on the signal during transmission, and it cannot accurately describe the mutual coupling effect between tags. And the instantaneous received power in Nakagami-*m* fading channel is then given by
(6)$$f_{p}\left(p\right)=\frac{m^{m}p^{m-1}}{{\bar{p}}^{m}\Gamma \left(m\right)}e^{-\frac{mp}{\bar{p}}}\left(p\geq 0,m\geq 0.5\right)$$
where *m* is the channel fading parameter, $\bar{p}$ is the average power of the tag’s signal, and $\Gamma \left(m\right)$ represents the Gamma function. This work was derived by adopting the Laplace and inverse Laplace transformation of the expressions for the PDF, but the details are omitted in this paper for brevity. Consequently, the compound PDF results from the convolution of n−1 PDFs of received power can be written as
(7)$$f_{p_{s-1}}\left(p_{s-1}\right)=\frac{m^{m\left(s-1\right)}}{\Gamma \left(ms-m\right){\bar{p}}^{m\left(s-1\right)}}p_{s-1}^{ms-m-1}e^{-\frac{mp_{s-1}}{\bar{p}}}.$$

Substituting (Equation 6) and (Equation 7) into (Equation 5), we can obtain [47]
(8)$$\alpha^{Nak}\left(s,q,m\right)=\frac{s}{\Gamma \left(m\right)\Gamma \left(ms-m\right)}\sum_{k=0}^{\infty }\frac{{\left(-1\right)}^{k}\Gamma \left(ms+k\right)}{k!\left(ms-m+k\right)q^{ms-m+k}}.$$

It can be seen from the above formula that the capture effect for the RFID system is not only affected by the capture threshold *q* and the channel parameter *m* but also by the number of tags *s* in the collision timeslots.

## 4. Capture-Aware Algorithm for Large-Scale Tag Identification

### 4.1. Proposed Estimation Algorithm

The capture-aware algorithm can achieve parameter estimation by counting the number of different timeslot states observed after each frame and further adjusting the frame length to improve the efficiency of tag identification. However, due to the lack of prior knowledge of the number of tags, especially during large-scale intensive tag identification, it is difficult for the initial frame length to match the true number of tags, which negatively affects the system’s efficiency of tag identification [29]. When the number of tags is much larger than the initial frame length, there are many collision timeslots in a frame, and tag identification efficiency is very low, and on the other hand, many idle timeslots can also affect the system’s identification delay. Therefore, we propose a method of solving the problem of matching the initial frame length to an unknown number of tags by using the first several timeslots in a frame. In this work, we define the probability that there are no idle timeslots in the first *w* timeslots, which can be expressed as
(9)$$P_{w}=1-\sum_{i=1}^{w}\left(\begin{array}{c} w\\ i \end{array}\right)p_{I}^{i}{\left(1-p_{I}\right)}^{w-i}$$
where the probability $p_{I}$ of the idle timeslot can be expressed as $a_{I}/l_{†}$. When $l_{†}$ = 128, theoretical and simulation results $P_{w}$ are given in Table 1, where *w* = 1 to 8 and the number of tags *n* varies from 50 to 1000.

The results from Table 1 indicate that the probability $P_{w}$ of having all non-empty timeslots in the first w timeslots increases with the number of tags regardless of the value of *w*. In the context of large-scale RFID tags, the initial frame length can be rapidly adjusted according to $l=\mu l_{†}$ when $P_{w}$ is close to 1. It is worth noting that $P_{w}$ being close to 1 is not an exact expression, and in experiments, we need to perform relevant debugging based on *w* and the parameter σ to determine the conditions for the rapid adjustment of the initial frame length, which will be introduced in Algorithm 1. Indeed, the estimation strategy of the proposed algorithm can be divided into two parts: the estimation of tags’ numbers and the estimation of capture probability. Then, considering that the number of idle timeslots observed by the reader is not affected by the capture effect of Nakagami-*m* fading channels, the number of tags can be estimated using idle timeslots. Invert $N_{i}=lp\left(0\right)$ and replace $N_{i}$ with the observed value $c_{i}^{cap}$ and the proposed algorithm estimates the number of tags by
(10)$$\hat{n}=\frac{\ln\left(c_{i}^{cap}/l\right)}{\ln\left(1-1/l\right)},c_{i}^{cap}\neq 0.$$

**Algorithm 1:** Pseudo-code operation of a reader.

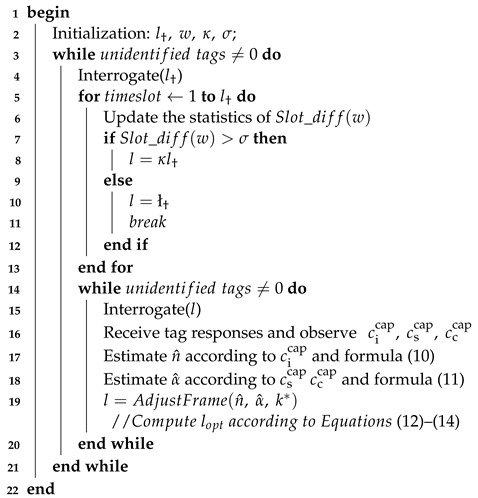



In addition, the number of unidentified tags in the i+1-th frame is $backlog=n_{est}-c_{s}$ if the estimated number of tags is $n_{est}$ after the *i*-th frame; then, the capture probability α can be shown to be
(11)$${\hat{\alpha }}_{cap}=arg\min_{\alpha_{cap}\in P}E\left\{{\left(c_{s}^{cap}-N_{s}^{{}^{\prime }}\right)}^{2}+{\left(c_{c}^{cap}-N_{c}^{{}^{\prime }}\right)}^{2}\right\}$$
based on the minimum mean square error. The proposed algorithm can accurately search the capture probability under the fading channel, where $c_{s}^{cap}$ and $c_{c}^{cap}$ denote the observed numbers of successful and collision timeslots, respectively; $N_{s}^{{}^{\prime }}$ and $N_{c}^{{}^{\prime }}$ denote their expectations, respectively; and *P* can be expressed as $P=\left\{\alpha_{cap}|0\leq \alpha_{cap}\leq 1\right\}$.

### 4.2. Frame Length Adjustment

In Section 2, it is observed that collision timeslots have the potential to convert into successful timeslots in the capture environment. However, setting a frame length that is too long may increase idle timeslots and ultimately lead to time slot waste. Conversely, reducing the frame length can increase the likelihood of capturing transmission slots but may also decrease system identification efficiency. Therefore, this paper proposes an optimal method for frame-length adjustment. Combining Equation (Equation 3) and capture-aware efficiency with the duration of different time slots can then be written as
(12)$$\eta_{id}=\frac{N_{s}t_{s}+N_{c}t_{c}\alpha_{cap}}{N_{i}t_{i}+N_{s}t_{s}+N_{c}t_{c}}$$
where $t_{i}$, $t_{s}$ and $t_{c}$ can be expressed as the duration of the idle time slot, the successful time slot, and the collision time slot, respectively. Divide the numerator and denominator of the above equation by $t_{s}$. Let $u=t_{i}/t_{s}$, $\upsilon =t_{c}/t_{s}$ and take a linear model ($l=kn,k\in R^{+}$) into account [48]. Combining the $\lim_{n\to \infty }{\left(1-1/kn\right)}^{n}=e^{-1/k}$, we can obtain
(13)$$\eta_{id}\approx \frac{\frac{1}{\left(ke^{\frac{1}{k}}+k-1\right)}+\alpha^{Nak}}{\frac{1+ku}{\left(ke^{\frac{1}{k}}-k-1\right)}+v+\left(1-v\right)\alpha^{Nak}}.$$

Then, the optimal frame length can be written by
(14)$$l_{opt}=\left[k^{*}\hat{n}\right]$$
where $k^{*}=arg\max_{k\in k}\left(\eta_{id}\right)$, k denotes the searching range of *k* and $\left[\cdot \right]$ is a floor integer function.

Table 2 shows the performance comparison between equal timeslots and unequal timeslots. According to the different frame size and system-throughput values corresponding to different channel shape parameters, we find that the method under an unequal time slot is better than the method under an equal time slot, which further verifies that the time slot setting in EPC C1 G2 protocol is still suitable under the influence of fading channel capture effect. Next, according to Equation (Equation 14), this work establishes the relationship between *n* and the optimal frame length $l_{opt}$ for different channel parameters *m* adopting the EPC C1 G2 standard, as depicted in Figure 5. It is evident that when the number of tags is constant, the RFID system requires a greater optimal frame length as *m* increases.

### 4.3. Description of the Algorithm

In the proposed algorithm, an identification cycle is depicted, which is implemented as shown in Algorithm 1.

To recognize densely distributed large-scale tags, the reader will send a query command with a short frame $l_{†}$ after initializing the parameters. Then, the tag will randomly select a time slot from the frame and respond to the query command. After that, the Slot_diff(w) and σ are compared, where Slot_diff(w) is used to record the calculation results of the first *w* slot states, and σ can control the pace of the frame length adjustment (*w* = 4, σ = 1 in this work). If the former is larger than the latter, a dynamic adjustment of the initial frame length *l* is performed according to the method of $l=\kappa l_{†}$. Otherwise, the adjustment is completed. After sending the query command with the frame length *l*, the reader will count the idle slots after each frame ends to estimate the number of tags. It will also count the number of successful slots and collision slots to estimate the capture probability. Finally, the optimal frame length is found according to the Formulas (Equation 12)–(Equation 14), and the reader will send a query command with the next frame length until all tags are recognized.

## 5. Numerical and Analytical Result

In this section, numerical results are presented to verify the performance of the proposed algorithm. The random data used in the experiments were generated using MATLAB’s internal function (MATLAB Version: 9.14.0.2194193 (R2023a)). Firstly, the capture effect was simulated using the Monte Carlo method, and 10,000 independent experiments were performed to obtain the average results. Simulation results, including Figure 6 and Figure 7, are presented to investigate the impact of various parameters on the capture performance of Nakagami-*m* fading channels in a dense tags environment. In the experiment, the numbers of collision slot tags were set to two, three, and four, and the theoretical value in the closed expression of capture probability in the Nakagami-*m* fading channel in Section 3 was compared with the actual value. Then, in order to more intuitively explore the influence of other factors on the capture effect, a 3D surface diagram of the capture probability under the Nakagami-*m* fading channel was presented in Figure 6. Additionally, in order to validate the performance and versatility of the proposed algorithm in estimating the capture probability, the capture probability under the actual Nakagami-*m* fading channel was not used. Instead, the search range for the capture probability was set to a larger range, $\alpha =\left\{0,\;0.1,\;0.2,\ldots ,1\right\}$, and the step size was 0.1. The prior distribution of the tag was set to be uniform over the range of μ, where μ varies from 50 to 1000. The search range for the tag of the CMEBE and MMSE, CBMS algorithms was set to $N=\left\{c_{s}^{cap}+2c_{c}^{cap}\leq n\leq N_{\max}|n\in Z\right\}$, where $N_{\max}=1000$. In the experiments, the initial frame length of the proposed algorithm was set to 128. The parameters *w*, κ, and σ were set to 4, 2, and 0.85 in the proposed algorithm, respectively. Based on the analysis in Table 2, the setting of time slot duration was $t_{i}$ = 50 μs, $t_{s}$ = 400 μs, and $t_{c}$ = 200 μs (u=0.125,v=0.5), which is consistent with the EPC C1 G2 Protocols, and the VOGT [49] algorithm used an equal duration for all timeslots. To evaluate the performance of an algorithm, the estimation error is defined as $error=\left|\frac{x-\hat{x}}{x}\right|\times 100\%$.

Figure 6 shows that when the capture threshold is constant, the capture effect is related to these two factors, which are the shape parameter *m* of the fading channel and the number of tags *s* in the collision timeslots. Among them, the value of the capture probability decreases with the increase in the shape parameter *m* of the channel; in other words, a larger *m* will make the capture effect more difficult to occur. The capture probability also decreases as the number of tags in the collision timeslots increases.

Figure 7 presents the same trend as Figure 6 and also shows that the capture probability is closely related to the coding and modulation capability of the reader. When the capture threshold is set to a large size, the conditions for satisfying the capture will be more stringent, such as q=7, m=2, and the capture probability is only 0.08 (s=2).

Afterward, we conducted simulations to evaluate the estimation performance of the proposed algorithm. The simulation results are shown in Figure 8 and Figure 9. In Figure 8, we compare the estimation error of the capture probability of the proposed algorithm with that of other algorithms. The results show that the estimation error of the capture probability of all four algorithms decreases as the capture probability increases. Compared with the other algorithms, the estimation algorithm for capture probability proposed in this paper demonstrates good estimation performance in the range of capture probabilities from 0 to 0.5. For example, when the capture probability is 0.2, the estimation error of the proposed algorithm is only 3.46%, much lower than that of the other three algorithms. Additionally, Figure 7 shows that the capture probability is usually small under a fading channel, which further demonstrates the superiority of the proposed algorithm.

Figure 9 illustrates the tag estimation performance of various algorithms in the presence of the capture effect. Notably, CMEBE and CAE exhibit suboptimal tag-estimation accuracy when the number of tags is large. Specifically, when there are more than 700 tags, CAE’s estimation algorithm becomes invalid due to a lack of free timeslots caused by mismatched initial frame length during tag estimation, which significantly impairs its performance. Because the VOGT algorithm does not have the capture sensing ability, it has a large estimation error when the number of tags exceeds 200. In contrast, both MMSE and CBMS and our proposed algorithm employ an identical initial frame-length-adjustment mechanism, resulting in similar performance trends for tag estimation. Simulation results indicate that with 1000 tags, the proposed algorithm achieves a tag estimation error of only 1.94%, which is a reduction of 29.96% compared to the MMSE, CBMS algorithm and 70.29% compared to the CAE algorithm. Therefore, this paper’s proposed algorithm exhibits superior tag estimation performance in the presence of a capture effect under fading channels.

The capture probability of a Nakagami-*m* fading channel is not only related to the encoding and modulation method of the reader but also depends on the degree of fading and the number of tags in the collision epoch. Therefore, compared to a fixed capture effect, we need to deeply explore the impact of different estimation algorithms on the stability of tag identification under the parameters of channel fading.

To simplify the analysis of the role of the capture effect in collision-avoidance strategies, we assume that the reader can count the number of tags in the collision epoch and use the formula, combined with the setting of the capture threshold, to obtain the probability of the occurrence of capture effect in different collision epochs under the fading channel. In this experiment, the number of tags is n=700, and the capture threshold is set to q=2.

When the capture threshold is fixed, the probability of the occurrence of the capture effect is mainly affected by the shape parameters of the fading channel. Therefore, different shape parameters will affect the efficiency of tag identification in the system. However, by combining with the channel environment, we can set the optimal next frame length to achieve the optimal overall efficiency of tag identification. To determine the next frame length in the VOGT algorithm, this article employs the method $l_{next}=\hat{n}-c_{s}^{cap}$. In contrast, for the CAE and CMEBE algorithms, method $l_{next}=\hat{\alpha }+\left(1-\hat{\alpha }\right)\left(\hat{n}-c_{s}^{cap}\right)$ is utilized to determine the subsequent frame length. As for both MMSE and CBMS and our proposed algorithm, we adjust the next frame length using the formulas (Equation 12)–(Equation 14).

Figure 10 shows the influence of shape parameters of the Nakagami-*m* fading channel on the tag-identification efficiency under different algorithms, in which channel shape parameters reflect the fading degree of the channel. When m=1, the fading channel follows Rayleigh distribution. It can be seen that the identification efficiency of the five algorithms decreases with the increase in the channel-shape parameter *m*, among which the VOGT algorithm has the worst tag-identification performance. When the shape parameter m=0.5, the proposed algorithm is very close to the tag-identification efficiency of CMEBE, CAE, MMSE, and CBMS, all reaching 83%, mainly because their estimation performance is not different when the capture probability is large. However, with the increase in the channel shape parameters, the capture effect becomes smaller. When m=3, the identification efficiency of the proposed algorithm is 76.83%, only declining by about six percentage points, indicating that the proposed algorithm exhibits strong stability compared to other algorithms.

Next, we employ the Big O notation to obtain the computational complexity of the proposed algorithm with that of pre-existing algorithms. Here, we specify that the set of estimation tags *n* and capture effect α of the above algorithm are *N* and *P*, respectively. Assuming that $\varepsilon =\left|P\right|$, $\varphi =\left|N\right|$, where $\left|\cdot \right|$ represents the cardinality of the set, the VOGT method needs to gradually search for an extreme value within a certain number of tags, and its computational complexity can be approximated as $O\left(\sum_{i=1}^{\varphi }{\tilde{n}}_{i}\right)$. CMEBE, MMSE, and CBMS all require a two-dimensional search for the number of tags and probability of capture effect. The computational complexity of CMEBE, MMSE, and CBMS can be approximated as $O\left(\varepsilon \sum_{i=1}^{\varphi }{\tilde{n}}_{i}\right)$, where ${\tilde{n}}_{i}$ represents the *i* search value of the number of tags, and CAE and the algorithms proposed in this paper involve one-dimensional searches, with the CAE algorithm having a complexity approximation of $O\left(1\right)$. The proposed algorithm has an estimated complexity of $O\left(\varepsilon \right)$.

Thus, based on the proposed model, a conclusion can be drawn from the experiment that the estimation performance and tag-identification efficiency of the proposed algorithm are better than those of other algorithms. Among them, Table 3 indicates that the proposed scheme possesses the features of low complexity and high identification efficiency in vehicular network tag identification.

## 6. Conclusions

In this paper, a Nakagami-*m* fading-channel-capture model is established based on vehicle cargo tags in a large-scale environment with dense tags, and an effective capture-aware estimation algorithm is proposed. Both theoretical analysis and simulation results have shown that the proposed capture-aware algorithm is superior to the reference algorithms. In addition, we derive the optimal frame length for different durations of slots, further confirming that the duration settings of slots in the EPC C1 G2 standard are also applicable in the channel capture model. Next, we will develop a reader equipped with capture-aware technology that is in line with the current level of hardware development and conduct real-world testing on dense vehicle tags or cargo tags to further enhance its performance. 

## Figures and Tables

**Figure 1 sensors-23-06792-f001:**
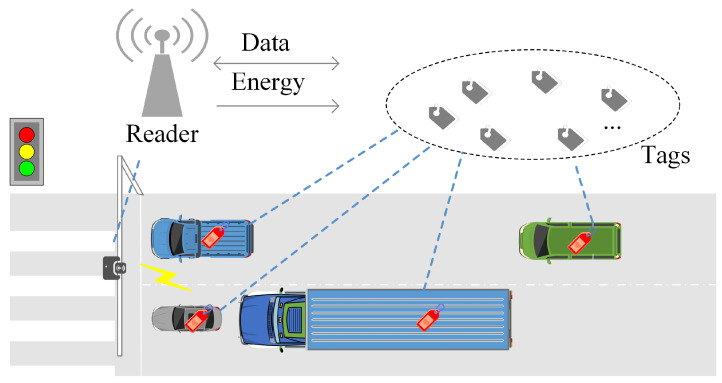
Vehicles with a high volume of cargo tags are statically detected at traffic intersections.

**Figure 2 sensors-23-06792-f002:**
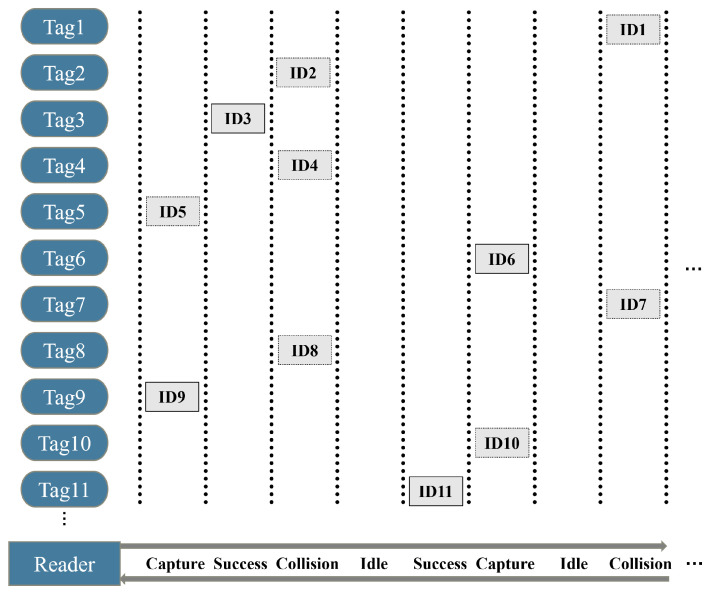
Detection results from ALOHA when the capture effect occurs.

**Figure 3 sensors-23-06792-f003:**
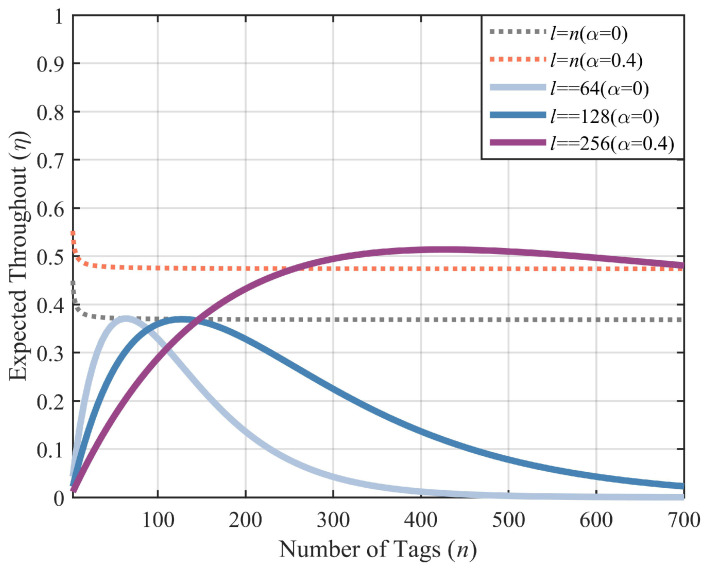
A comparison of FSA and DFSA with the capture effect.

**Figure 4 sensors-23-06792-f004:**
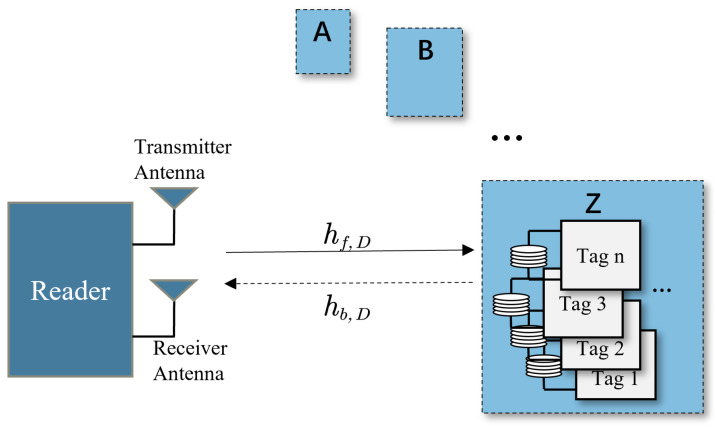
A monostatic system with transmit and receive antennas to be colocated at the reader (the reader device is positioned on the left side of the diagram, while the dotted line box on the right represents areas with high-density tag distribution).

**Figure 5 sensors-23-06792-f005:**
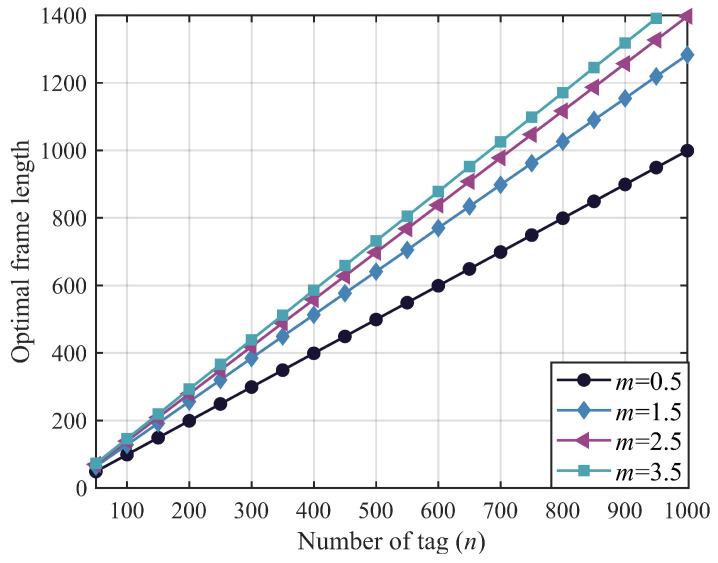
Optimal frame length $l_{opt}$ as function of tags *n* (q=2,u=0.125,v=0.5).

**Figure 6 sensors-23-06792-f006:**
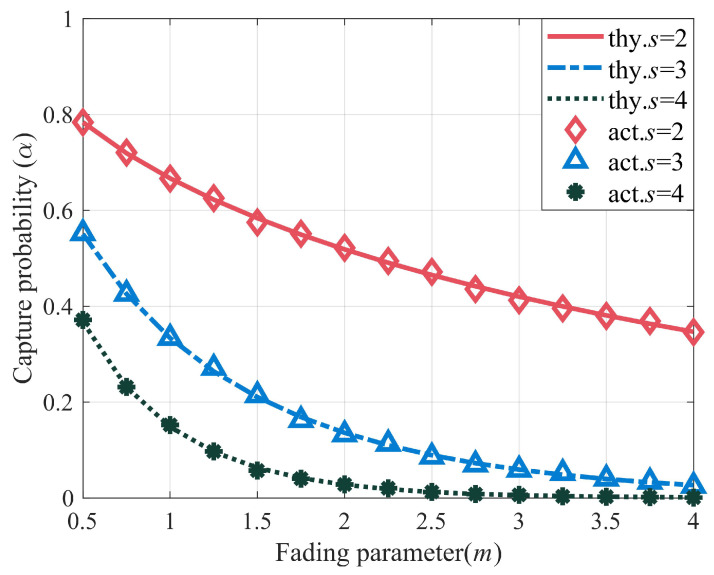
Capture probability on the Nakagami-*m* fading channel (q=2).

**Figure 7 sensors-23-06792-f007:**
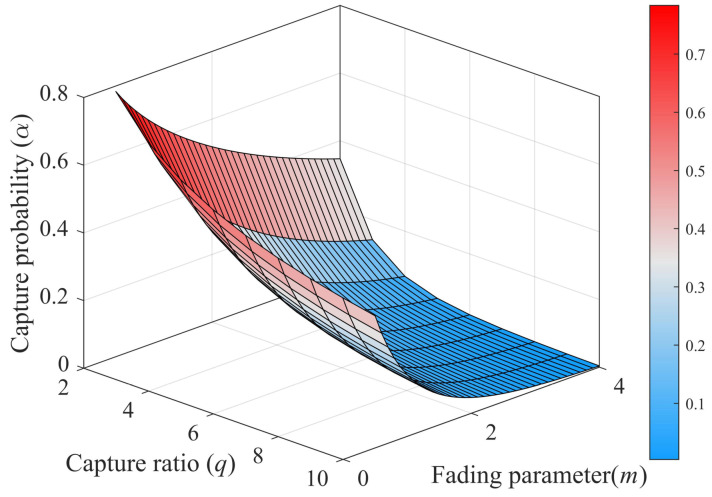
3D Capture probability on Nakagami-*m* fading (s=2).

**Figure 8 sensors-23-06792-f008:**
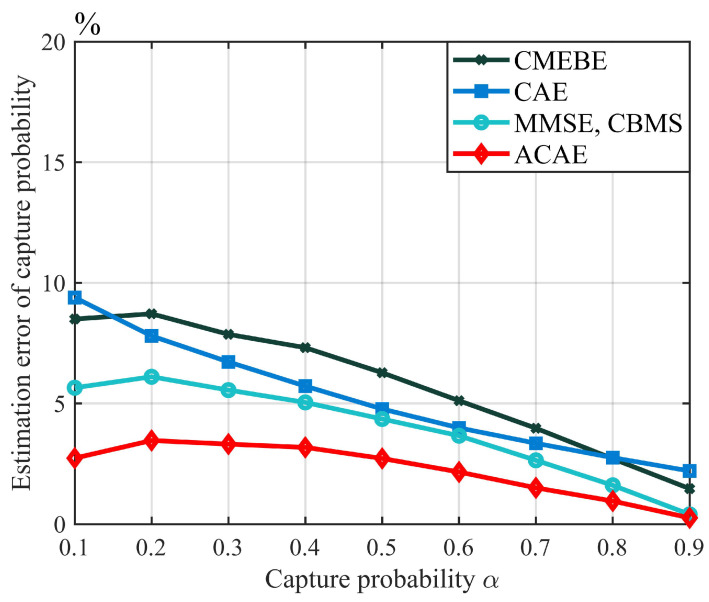
Capture probability estimation of the proposed algorithm (l=128, *n* = 700).

**Figure 9 sensors-23-06792-f009:**
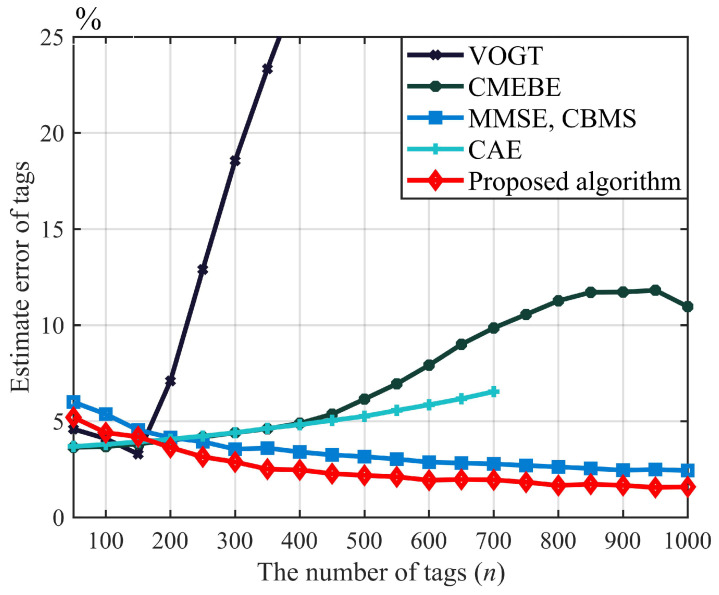
Tag estimation of proposed algorithm under fading environment (*m* = 1.5, *q* = 2).

**Figure 10 sensors-23-06792-f010:**
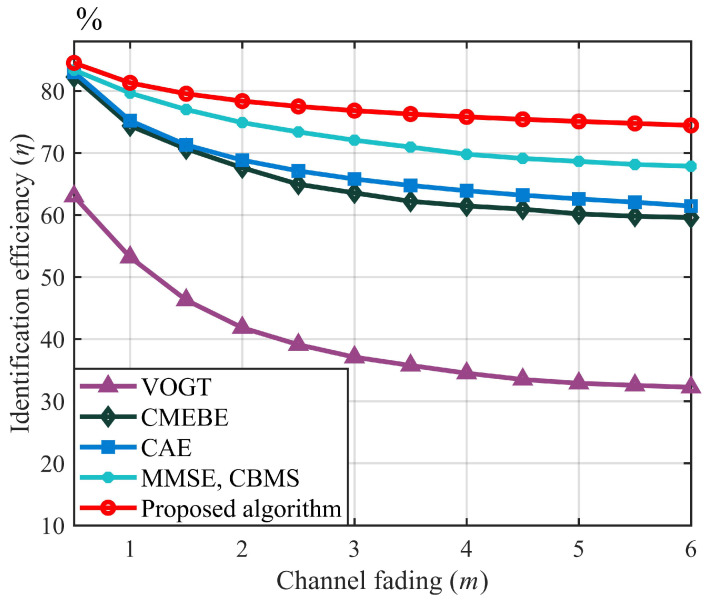
Comparison of the identification efficiency of five algorithms.

**Table 1 sensors-23-06792-t001:** Probability of no idle slot in the first *w* slots when $l_{†}$ = 128.

*n*	w=1	w=2	w=4	w=6	w=8
50	0.3244	0.1052	0.0110	0.0011	0.0001
	(0.3241)	(0.1055)	(0.0110)	(0.0010)	(0)
100	0.5436	0.2955	0.0873	0.0258	0.0076
	(0.5439)	(0.2956)	(0.0871)	(0.0257)	(0.0074)
400	0.9566	0.9151	0.8374	0.7663	0.7012
	(0.9557)	(0.9152)	(0.8374)	(0.7675)	(0.7010)
700	0.9959	0.9918	0.9836	0.9755	0.9675
	(0.9956)	(0.9905)	(0.9833)	(0.9750)	(0.9677)
1000	0.9996	0.9992	0.9984	0.9976	0.9969
	(1.0000)	(0.9995)	(0.9983)	(0.9977)	(0.9966)

where the results given in the brackets are computed from (Equation 9).

**Table 2 sensors-23-06792-t002:** The values of $l_{opt}$ and $\eta_{id}$ when *n* = 700.

Fading Channels (*m*)	u=v=1			u=0.125,v=0.5	
	$l_{\mathrm{opt}}$	$\eta_{\mathrm{id}}$		$l_{\mathrm{opt}}$	$\eta_{\mathrm{id}}$
0.5	393	0.6303		699	0.8427
1	466	0.5522		828	0.8116
1.5	508	0.5131		898	0.7938
2	535	0.4885		944	0.7823
2.5	555	0.4712		978	0.7738
3	571	0.4581		1004	0.7672
3.5	583	0.4478		1025	0.7618

where the results are computed from the Equation (Equation 12)–(Equation 14).

**Table 3 sensors-23-06792-t003:** Comparison of estimation methods.

Estimation Method	Estimated Object	Number of Identification Tags	Estimation Error of Capture Probability (%)	Estimation Error of Tags (%)	Identification Efficiency (%)	Computational Complexity	Run Time (s)
VOGT	$\hat{n}$	Small	-	46.74	54.21	$O\left(\sum_{i=1}^{\varphi }{\tilde{n}}_{i}\right)$	5.7462
CMEBE	$\langle \hat{n},\hat{\alpha }\rangle$	Small	8.50	9.84	72.86	$O\left(\varepsilon \sum_{i=1}^{\varphi }{\tilde{n}}_{i}\right)$	14.3671
CAE	$\langle \hat{n},\hat{\alpha }\rangle$	Small	9.38	6.53	73.21	$O\left(1\right)$	1.15
MMSE, CBMS	$\langle \hat{n},\hat{\alpha }\rangle$	Large	5.65	2.77	79.66	$O\left(\varepsilon \sum_{i=1}^{\varphi }{\tilde{n}}_{i}\right)$	32.61/33.13
Proposed algorithm	$\langle \hat{n},\hat{\alpha }\rangle$	Large	2.73	1.94	81.31	$O\left(\varepsilon \right)$	15.42

where the estimation error of capture probability when α=0.2 and the estimation error of tags and identification efficiency when n=700.

## Data Availability

The data that support the findings of this study are available from the corresponding authors upon reasonable request.

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
