# Peer review of "Capture-Aware Dense Tag Identification Using RFID Systems in Vehicular Networks"

_sensors, 2023, doi:10.3390/s23156792_

Round 1

Reviewer 1 Report

The proposed work brings a solution to the problem of overload in roadside readers in Passive Radio-Frequency Identification Systems. This problem happens when vehicles transmit tags to fixed equipment in the roadside but the node density makes the communication impracticable. For this, the article introduces an algorithm to improve efficiency in tag identification, using Nakagami-m distribution. The simulations manifest a great performance of the proposal, outperforming alternatives in the literature.

In general lines, the article is well-structured and correctly written. Next, I include some little considerations to improve the work.

- The conclusion section includes a table which summarises the results. This is not very common and I think the explanation about the table and the table would be much better located at the end of the result section.

Considering the approach and its scientific contribution, I recommend the publication of the work.

Author Response

Thank you very much for your valuable feedback! Kindly refer to the attached document to view our response.

Reviewer 2 Report

The posed questions do not in any way undermine the value and relevance of the study's findings. Rather, they serve to further validate the soundness of the research. Upon addressing these questions, this paper is well-positioned for publication, adding considerable value to the field of RFID tag identification in densely distributed vehicular networks.

Author Response

(The authors gave the same response as above.)

Reviewer 3 Report

I rate the reviewed article very highly. The authors have clearly presented the basics of passive radio-frequency identification (RFID) systems and indicated the problems with their use. I find the proposed way of solving the problems encountered interesting and possible to apply in real conditions. I wish the Authors successful road tests.

I believe that the article should be published immediately in the journal Sensors.

Author Response

(The authors gave the same response as above.)
